# Leading Diverse Workforces: Perspectives from Managers and Employers about Dyslexic Employees in Australian Workplaces

**DOI:** 10.3390/ijerph191911991

**Published:** 2022-09-22

**Authors:** Shae Wissell, Leila Karimi, Tanya Serry, Lisa Furlong, Judith Hudson

**Affiliations:** 1School of Psychology and Public Health, La Trobe University, Bundoora, VIC 3086, Australia; 2School of Health and Biomedical Sciences, RMIT University, Melbourne, VIC 3001, Australia; 3School of Medicine and Healthcare Management, Caucasus University, Tbilisi 0102, Georgia; 4School of Education, La Trobe University, Bundoora, VIC 3086, Australia; 5Child Well-Being Research Institute, University of Canterbury, Christchurch 8140, New Zealand; 6School of Education, University Tasmania, Hobart, TAS 7000, Australia

**Keywords:** dyslexia, workplace, employers, managers

## Abstract

Background: Dyslexia is a specific learning disability affecting around 1 in 10 Australian adults. It presents unique challenges for employees in the workforce, yet community and workplace awareness of the challenges of dyslexia is limited. The aim of this preliminary research was to explore the experiences and perspectives of Australian employers and managers responsible for supervising employees with dyslexia in the workplace. Materials and Methods: Using a qualitative research design, we conducted in-depth interviews with four managers who had current or previous experience managing employees with dyslexia. We used a deductive approach to analyse the data and categorise responses to the study questions. Results: Participant responses indicated that there is a lack of awareness and understanding of dyslexia within Australian workplaces. Participants identified challenges facing employees with dyslexia in the workplace including, differing personal levels of confidence and comfort in disclosing disability; the possibility of discrimination, and a lack of inclusive organisational practices and processes. Suggestions for ways to improve workplaces for dyslexic employees included: additional support for leaders and managers to drive inclusive leadership, and additional training for leaders and managers on how to best support employees with dyslexia. Conclusions: While only a small sample size, this study indicates that further research is needed to better understand the working environment of Australian leaders and managers. It appears that leaders and mangers need skills and knowledge to better support employees with dyslexia and in doing so create more inclusive workplaces.

## 1. Background

Dyslexia is the most common type of Specific Learning Disability (SLD) and affects the capacity of individuals to learn, to read and to write [1,2,3]. These skills are impacted by deficits in phonological and associated orthographic processing skills. Difficulties in decoding individual words has the greatest impact on reading [2,4,5]. Reported estimates of the prevalence of dyslexia varies due to diagnostic criteria but ranges from 5% to 20% [6,7,8,9,10]. Dyslexia occurs across the spectrum of intellectual ability [11,12,13]. Many individuals with dyslexia do become readers, although the route to achieving this is typically much slower and requires a significant amount of intervention from suitably trained practitioners [2,14]. In addition, because dyslexia is a lifelong condition [2,15], reading (along with spelling and writing) remains somewhat effortful and less automatic for people with dyslexia compared to their non-dyslexic peers [4,16]. Approximately 3–7% of the population have dyscalculia [17,18,19,20,21] have severe difficulties performing arithmetic calculations that persist into adulthood [17,18,19,20,21]. While approximately 3–15% of the population have dysgraphia, a specific learning disorder affecting the written expression of symbols and words that persist into adulthood [22,23,24].

Australian and international research has identified that people with dyslexia work across a range of sectors, and skill levels [25,26,27,28,29,30]. Accordingly, many employers may find themselves in the position of supervising employees with dyslexia in their workplace. Given the expected trajectory of difficulties related to reading, spelling, and writing, individuals with dyslexia are at increased risk of not meeting workplace demands or timeframes, despite having the training and/or qualifications to do so [30]. This is particularly evident if there is limited provision of suitable accommodations in the workplace [31,32,33,34,35].

There also remains a limited understanding of the impact of dyslexia in the broader community [36], and this is likely to be reflected in Australian workplaces [26,37]. For example, various authors, including Thorpe and Burns (2016) [38] and O’Dwyer and Thorpe (2013) [39] have identified that employers and managers may not have access to quality training about working with staff who have dyslexia. As such, dyslexia is still an invisible or ‘hidden’ disability within the workforce, with dyslexic employees bearing the brunt of the responsibility to disclose their disability and self-advocate for support. Several studies have found limited policies and procedures in place to support managers of teams that include dyslexic staff.

Recently the term ‘disability’ has also broadened to encompass both visible (e.g., physical limitations) and invisible (e.g., chronic fatigue, attention deficit hyperactivity disorder) conditions [40,41]. Along with the broadening of the term ‘disability’ there has been a cultural shift in global and national policy regarding the need to increase community awareness of disability and inclusion as seen through the United Nations Convention on the Rights of Persons with Disability in 2006 which Australia is a signatory of, and as such has a responsibility to protect the human rights and inherent dignity of persons with disabilities [42]. More poignantly, Article 27—Work and employment that recognised *the “ right of persons with disabilities to work, on an equal basis with others; this includes the right to the opportunity to gain a living by work freely chosen or accepted in a labour market and work environment that is open, inclusive and accessible to persons with disabilities”* [43]. At a national level Australians have seen initiative such as the Australia’s National Disability Strategy, implemented in 2010 [44] and expanded in 2021 and more recently the National Disability Insurance Scheme, which provides “*support to people with intellectual, physical, sensory, cognitive and psychosocial disability”* [45]. The initiatives outline Australia’s national aspirations to respect inclusivity and equality and outlines state government and ‘whole of community’ responsibilities [44]. Change towards greater inclusivity is also endorsed in legislative requirements such as the Discrimination, Equality and Fair Work Acts [46,47,48]. This makes clear that workplaces need to intensify their focus on diversity and inclusion for all people, particularly minority groups and those more likely to experience discrimination relating to sexual orientation, cultural background, age and gender, and disability [49,50,51,52].

However, despite the rollout of progressive international and national strategies [44,45] and changes to legislation [47,48] individuals with dyslexia have been left behind in particular within the employment sector as highlighted by research undertaken by Wissell et al. (2022) [30] who found those with dyslexia faced a significant number of barriers to succeeding in the workplace despite these legislations and national strategies. In fact, those with learning disabilities have been excluded from the National Disability Insurance Scheme (NDIS) and the Medicare Benefit Scheme (MBS) an Australia medical healthcare services subside scheme [53].

Considering these global, national policies and more inclusive and flexible workplace trends, employers are now expected to hire and support a diverse workforce [54]. However, there is minimal research to date investigating the benefits and challenges of managing a diverse workforce with a focus on dyslexia in the Australian context. As society demands more inclusivity, including from within the workplace, employers, managers and leaders will need to consider how best to incorporate dyslexia into their inclusion and diversity frameworks to provide a truly safe and inclusive workplace for employees of all abilities.

## 2. Aim

This research aimed to explore the experiences and perspectives of Australian employers and managers responsible for supervising employees with dyslexia in the workplace. This is a preliminary study that forms part of a larger research project that investigated the lived experiences of adults with dyslexia in Australia, focusing on the workplace.

## 3. Method

A qualitative design was selected for this study to explore individuals’ experiences of employing and managing people with dyslexia in Australian workplaces. We adopted a deductive content analysis approach using a set of pre-defined questions conducted through in-depth interviews to collect substantial responses from participants enabling us to draw specific conclusions from the data. Further follow-up and clarifying questions emerged during interviews with participants. Unlike quantitative research, qualitative research gathers data until saturation is reached, and the collected data tell a rich, complex, and in-depth story about the topic under examination. Compared to quantitative research, sample sizes for qualitative research are typically much smaller [55,56]. It has been suggested in a large number of articles, book chapters, and books that anywhere from five to fifty participants is sufficient [56].

### 3.1. Participants

Purposive sampling was used to recruit four participants. This method was chosen to select participants who could provide a detailed understanding of the experience of managing and working with employees with dyslexia [57]. Inclusion criteria required that participants: (i) worked in Australia within an Australian-owned company to keep the research focus within the Australian context, (ii) held a management position and/or position at a leadership level (iii) had a minimum of 10 employees within their organisation. It would be expected that with over 10 employees one would be dyslexic based on prevalence rates (iv) had worked in that organisation for more than three years and (v) had managed staff that had disclosed they had dyslexia. Ethics approval was received from [redacted University] to conduct this study.

Four eligible participants (three female and one male) agreed to participate. Table 1 displays their characteristics. Two participants were business owners, and two participants held senior leadership positions. The size of each organisation ranged from 13 to more than 1000 employees.

Three of the four participants had at least one other family member with dyslexia (a parent and/or a parent and a sibling). The participant with a non-familial dyslexic background had been educated about dyslexia via a formal workplace training and assessment course early in their career.

#### Data Collection

Author one, held one structured in-depth interview with each participant via online platform communication Zoom^®^. A purpose-designed set of questions was used to elicit participants’ views and experiences of leading or managing work colleagues with dyslexia (Appendix A). Questions were drawn from the literature about disability, in general, and dyslexia, more specifically, in the workplace [52,58,59] and from the lived experiences of the first author who has dyslexia. Interviews ranged in length from 30–107 min (mean = 54 min). Author One positioned herself within the context of this study by disclosing her dyslexia at the beginning of each interview. This was done to acknowledge and establish the authenticity, transparency, and context of the research [60,61,62].

The interviews were assigned a number code, and transcripts were deidentified by the first author and then sent to an external company to be transcribed verbatim prior to analysis. The interviewer kept field notes and conducted reflective journaling following each interview to assist in building a rich understanding of participants’ experiences in the context of the phenomena under review [63,64].

Prior to data analysis, each participant received their transcript and was invited to review and request changes to any part of the transcript within two weeks of receipt. No amendments were requested. Following this, transcripts were uploaded to NVIVO software (version 12).

### 3.2. Analysis

Deductive content analysis was used to analyse the data [65]. The choice of this analytical approach was based on the nature of the data; that is, data collected via structured interviews.

Over a three-phase process: preparation (phase one), organisation (phase two), and reporting (phase three), the data were sorted into several categories [65]. Phase one initially involved familiarisation with the interview data through multiple active readings of the transcripts and listening to audio files. This resulted in the identification of two main categories: (1) Employees with dyslexia and their workplace challenges and (2) Workplace enablers that support dyslexic employees.

During phase two, text from the interview transcripts were sorted into the two main categories developed in phase one. Following this, further analysis was conducted. Descriptive open coding, involving a search of the data for high-frequency words, phrases, and sentences (e.g., ‘very skilled, but she was a little bit slower to understand briefs’), was undertaken to identify critical features and nuances in the data. These open codes and high-frequency words, phrases, and sentences formed sub-categories of data. For example, the sub-category ‘Workplace performance’ was identified under the main category identified in phase one, ‘Employees with dyslexia and their workplace challenges’.

Transcripts were then re-read for reflection and to ensure that interpretation of participants’ data had been captured authentically. The data was condensed further to create five sub-categories associated with the two main categories. Direct quotations from participants were cross-checked against the categories to ensure the categories were reflective of the participants’ narrative [66,67]. Although the researcher had pre-set discussion topics within the interviews, the main categories and sub-categories were created from the interview data. Finally, phase three consisted of writing up the report.

## 4. Results

The two main categories and five sub-categories identified from the data. The results highlight the myriad of challenges faced by employees with dyslexia, but also provide examples of how employers and managers could better support dyslexic employees.

### 4.1. Employees with Dyslexia and Workplace Challenges

This first main category from the interview data focused on the challenges of working with and providing leadership to dyslexic employees. Participant responses were categorised as follows: (1) employers and managers perceived dyslexic employees’ workplace performance; (2) self-disclosure of disability by dyslexic employees and (3) perceived discrimination in the workplace, a manager’s perspective.

#### 4.1.1. Dyslexic Employees’ Workplace Performance

All participants felt that dyslexia could be a barrier in the workplace regarding dyslexic employees executing their expected administrative and professional duties. For example, participants had observed employees with dyslexia having difficulties constructing emails, problems with spelling and grammar, delayed processing of written information and needing extra time to complete tasks that involved reading and writing. These difficulties were prominent when participants were asked to compare dyslexic employees to those without dyslexia.

Prior to knowing that employees had dyslexia, employers and managers noted they had often questioned the reasons for minor task errors and the length of time required to accomplish tasks. For example, P3 described how during a performance review, it had been noted that his employee was:


*“…a little bit slower than the other staff and sort of needed to pick up the pace and she needed to work on the interpretation of briefs a little bit better…”*


During this same performance review, the employee disclosed they had dyslexia. Following their disclosure, P3 reported that his attitude towards his employee’s work shifted, and strategies were put in place to allow the employee more time to complete tasks. More detailed instructions were also verbally provided for future task briefings.

Some workplace challenges described by participants were unique to the characteristics of the workplace and sector. Within the hospitality industry, these challenges included; matching invoices to product numbers, writing down phone numbers and people’s names, interpreting written information and communicating with individuals who had English as a second language.


*“Sometimes we’ve got these big numbers, which are product codes or item numbers for reference. And then if the box says ‘food’, and the food item is written in its long form, but the box perhaps has the short form, they can’t determine that that is that item, because… you know, it doesn’t match.”*
[P1]

In the not-for-profit sector, challenges included writing briefs, preparing tenders, and preparing government documents. In the construction and manufacturing industry, performance challenges were evidence for dyslexic employees in management roles, particularly when these managers were required to provide instructions for team members in written format (such as step-by-step instruction manuals) or when they were required to work at a fast pace.

#### 4.1.2. Self-Disclosure of Disability by Dyslexic Employees

Of the four participants, two had worked with multiple dyslexic employees, whilst the other two had only worked with one staff member who had disclosed dyslexia. Participants identified several barriers for employees to disclose in the workplace.

In most instances, they reported that employees disclosed only once a manager identified areas of difficulty, poor performance or a failure to meet expectations. In other instances, disclosure occurred when employees had passed their probation period or following a good performance review.


*“…One team member had joined us, and had passed their probation period, [but needed to] develop their written communication [skills]. When he came to me, he was quite open and said ‘Look, I do have a lot of difficulties in this area and this is my concern, which is my problem, my disability…”*
[P1]

Participants perceived those employees with dyslexia may fear discrimination following disclosure. In most cases, participants reported that their employees appeared embarrassed, anxious, ashamed, and lacked the confidence to self-advocate following disclosure. Some felt that employees might have had previous negative experiences when they disclosed their dyslexia to a manager and feared possible repercussions.


*“…[They] told me after a very good review that [they] had dyslexia. They just said there’s something else that they need to tell me, and I could see them start to tremble, and the anxiety in their face and fear. Judging from that, I would expect employers may have discriminated against people with a learning disability, or possibly managed it in a way that wasn’t so proactive and positive for the employee…”*
[P3]

Even after employees had disclosed their dyslexia, workplace policies and procedures were not adjusted to make the workplace more inclusive.


*“…We do have a non-discriminatory policy, obviously. It’s more geared towards ‘this is the legislation, and this is what you need to be aware of.’ It’s not an individual sort of “this is the way we want you to handle it if (sic) [it’s] brought up, or if you become aware of it.” So, no, there isn’t a specific policy that deals with it at all…”*
[P1]

Participants 1, 2 and 3 all felt another barrier to disclosure was when managers and employers were unskilled in dealing with disclosure. The participants also felt that due to a lack of understanding of dyslexia across organisations, other managers would not necessarily have the skills to support dyslexic employees, which could lead to discrimination and conflict. In two instances, staff members had not disclosed their dyslexia until performance management was put into effect.


*“…In two instances the employee was brought to HR. These people had got along [with their teams], they had worked out ways of working that worked [for them] the people that knew them well, knew their contribution, and valued them highly. However, then somebody came along and asked for something different of them, that wasn’t their core-strength, and they weren’t able to do it. They just weren’t confident to either disclose the dyslexia or manage the situation in a way to get the other person off their back. And it turned into a conflict…”*
[P4]

#### 4.1.3. Discrimination in the Workplace, a Manager’s Perspective

Participants felt there was a considerable lack of awareness and understanding of how to manage and support individuals with learning disabilities at an organisational and leadership level, which they felt could impede an employee’s ability to succeed in the workplace. They felt this lack of awareness and understanding could also possibly explain an employee’s fear of discrimination from work colleagues and organisations.

Overall, participants thought there should be a better understanding of dyslexia in the workplace, including both the strengths and weaknesses of individuals with dyslexia and ways to reduce the stigma and discrimination that dyslexic employees could experience. Participants felt this might prevent situations like the following example provided by P4, who spoke of an employee in the role of team leader:


*“…The team leader would not provide the information that they [their staff] required in a written format. The staff felt the team leader was deliberately holding them up from progressing in their career. The team leader was about to be stood down pending further investigation. He broke down, and disclosed he had dyslexia, which he had been able to keep undisclosed from the company. He said he didn’t want anyone to know. He felt [a lot of] shame and was willing to take the disciplinary action as a choice over disclosing the dyslexia. That’s how strongly he felt about it…”*


Participants observed a mix of attitudes between non-dyslexic work colleagues and work colleagues with dyslexia. Some observed positive responses such as informal support between team members and constructive responses of team members to the disclosure of dyslexia. However, some participants described seeing “*direct discrimination*” [P1, P2, P3] against those with dyslexia in the workplace, including dyslexics being called “*dumb*” [P4] and “*stupid*” [P4] and colleagues “*sniggering*” [P4] about people’s spelling mistakes. They described how they could imagine employees being undermined and their contributions not valued because of their poor written communication skills.


*“… If you get a document that you can tell is just not up to a standard, I can imagine in an environment where no one knows anything [about dyslexia]…, I could imagine very easily that you’d get bullied because a person would just say you’re not doing your job, you’re doing a crap job, what is this work, it’s not enough, it’s not what I expect, and you’ve got to do better…”*
[P2]

P1 described a view that there was still a stigma attached to the term *dyslexia*, which prevented people from disclosing their dyslexia or disclosing dyslexia as a disability. 


*“…There needs to be the stigma removal of dyslexia because I think they’re very cautious about sharing with others in case they’re treated differently…”*
[P1]

Participants also believed that dyslexia awareness training was needed at both a team and leadership level.


*“…I think we could always do with education of others. If you’ve got team members that are dyslexic, there are obviously concerns, there are considerations, and this is what we should and shouldn’t do. But I think we need to educate the rest of the team first. Because otherwise you’ll have some people that will think that, you know, they’re getting a better run or they don’t have to do all of this, and we’re all getting paid the same money. There’s always going to be that aspect…”*
[P4]

#### 4.1.4. Organisational Practices, Policies and Processes

Three participants felt that poor workplace practices and organisational processes could lead to unintentional discrimination of dyslexic employees. They described barriers to employees accessing appropriate support across the employment life cycle, from the early stages of recruitment through to retention of staff. Recruitment and induction procedures such as addressing selection criteria and pre-employment tests were seen as barriers to employment where those with dyslexia could not openly disclose their dyslexia and/or request access to the support they were entitled to. Additionally, participants observed that onboarding processes requiring large amounts of reading could alienate potential or new employees, leaving them at risk of not understanding workplace roles and responsibilities while also exposing companies to liability.


*“…Shortlisted candidates have to do psychometric testing, which is online testing. There’s a lot of reading in the tests. It’s not easy. I can imagine that is a barrier for someone with severe dyslexia. And then there’s quite a lot of reading [going forward]. There are heaps of policies, there’s heaps of documents to read, heaps of readings, essentially. So that’s also not that helpful…”*
[P2]

Participants identified a lack of experience of organisational leaders to manage and support individuals with disabilities and differing staff attitudes towards those with disabilities. They described a need for workplaces to be more inclusive. Suggestions as to what measures could be adopted included employing more people with disabilities and employing more neurodiverse individuals. Participants felt this would progress the implementation of inclusion and diversity committees, clear policies, and qualified human resources managers adequately trained to deal with various disabilities in a more diverse staff. However, for the organisations represented through our interviews, only two organisations had implemented strategies to support staff with disabilities, with little focus on dyslexic individuals.


*“…I think there is a growing awareness in our society about neurodiversity and all the value that brings. Personally, I think in the last five years, maybe even 10, there’s a heap more awareness. It’s an opportune time for there to be a rethink in terms of what we mean by inclusion…”*
[P2]

### 4.2. Workplace Enablers That Support Dyslexic Employees

The second main category from the interview data focused on the workplace enablers that can support employees with dyslexia to work to their full potential and meet employer expectations. Results from this category were sorted into two sub-categories: (1) inclusive leadership, and (2) workplace training.

#### 4.2.1. Inclusive Leadership in Action to Support Individuals with Dyslexia in the Workplace

Participants indicated that even with limited workplace policies and procedures in place, those with dyslexia could thrive if their employers and managers are responsive, empathetic, and upskilled to work with them. Those participants with a personal connection to dyslexia, and an awareness of the strengths that a dyslexic individual can possess, expressed a sense of positivity in working with dyslexic colleagues. They also expressed that implementing reasonable adjustments to support employees with dyslexia was no more burdensome than meeting the requirements of other non-disabled or non-dyslexic staff. Further, they noted that most adjustments they perceived to be effective were free, or low cost.

All participants felt equipped and confident to support their staff and to make accommodations, despite there being no formal procedures in place in their organisation, due to their personal encounters with dyslexia “*...It was just giving her more time to be able to clearly go through things, rather than trying to go at a pace that wasn‘t beneficial to her understanding or checking or doing everything correctly...*” [P3].

Participants reported that some accommodations were made in collaboration with their dyslexic employees, while other accommodations were implemented based on role requirements. Accommodations included a buddy system, meeting more frequently for one-on-one informal support, allowing more time to complete tasks, providing instructions in a verbal format, encouraging open communication, and providing ongoing verbal feedback.


*“…I always think about who she [the employee] should work with when we’re doing big pieces of work, like big reports, in that I know that if I assign her a project officer to work alongside her to check everything and to do the editing and all of that stuff, then I know it’s going to be a good end result…”*
[P2]

#### 4.2.2. Workplace Training to Increase Understanding and Awareness of Dyslexia

Our participants identified that leading and managing teams of people with different needs can be complex, especially for leaders that have a limited understanding of these needs. Participants suggested that a barrier to effectively leading a diverse team could be the little understanding and lack of training that managers and leaders have about dyslexia.


*“…I think that managers see it [dyslexia] as something that’s difficult to deal with. It makes their job harder. Everyone’s time is precious, so unless they’ve got an open mind, I think sometimes it’s seen as something that’s a little bit hard to deal with, and that’s going to add another element to their already busy day. Workplaces don’t have enough knowledge of [dyslexia]. Having to stretch your mind to trying to understand LGBTQ, then multifaith, multicultural. Now we’ve got neurodiverse thinking and then you’ve got dyslexia, autism, anxiety, depression, mental health. It’s like “I don’t have a psychology degree, how am I meant to manage all of this?”*
[P4]

Therefore, workplace training was identified as an enabler for better supporting employees with dyslexia, their teams, their managers, and organisations. It was noted that workplace training was a way of reducing the stigma attached to dyslexia and discrimination within the workplace. Only one of the four participants had received formal training that had included information about dyslexia.

All participants noted that when a staff member disclosed their dyslexia, there were no resources available inside their organisations to support them as managers, and they also had difficulty locating appropriate resources outside of their organisations. They felt Australian-led dyslexic organisations provided support targeted at children rather than adults. All participants felt that training on awareness and available resources was imperative to improve how individuals with dyslexia were supported and general workplace culture.


*“…I think it would be good for people to have more awareness around the link between reasonable adjustments [in the workplace] and dyslexia. Because that is sort of an advocacy movement. So, it potentially could be an awareness program that would be very helpful for people that have dyslexia…”*
[P4]

## 5. Discussion

In this study, we adopted a deductive content analysis approach to explore employers’ and managers’ perspectives and experiences of leading and managing diverse teams, particularly employees with dyslexia. Currently, there is minimal literature investigating how best to manage a diverse workforce, particularly with a focus on dyslexia. This was surprising given the high proportion of people with dyslexia in the population. However, we do know there is a significant amount of international literature on the workplace experiences of dyslexic employees [26,34,37,68] and these outcomes align with and support the employer and managers’ experiences and perceptions described and represented in our findings. Local [27,69] and international studies [25,26,28] suggest that individuals with dyslexia work across a variety of occupations and roles. Therefore, people in leadership roles across all industries will most likely encounter and manage dyslexic individuals at some point in their careers.

### 5.1. Workplace Performance of Dyslexic Employees

Although based on a small sample, our findings suggest that the nature of dyslexia may pose unique challenges for employers and managers. Our participants identified several literacy-based difficulties at a task level faced by dyslexic employees, including errors with spelling and grammar, length of time to complete work tasks, poor organisation and time management, and difficulties with reading comprehension and speed. These findings correlate with several international studies focusing on the experiences of dyslexic employees. Sumner and Brown (2015) [41] found that employees with dyslexia face unique challenges because the nature of their disability directly affects various work practices. In addition to the challenges that are common to all people, such as increased cognitive demand and an ability to respond in fast-paced work environments, employees with dyslexia must also contend with their literacy-based difficulties and this has the potential to put employees with dyslexia at a considerable disadvantage within the workplace [41,70,71,72].

Our findings also suggested that the impact of dyslexia may not become apparent until work performance drops. Although our results are based on a small sample size, they do support the work of De Beer et al. (2014) [26] and Winters (2020) [73], who found that managers may only suspect a problem when they see employees unable to keep up with speed of completing tasks or not coping with significant changes (e.g., new work procedures, managers, team or equipment). This suggests that workplaces are currently reactive, rather than proactive, in their awareness and support of disabilities such as dyslexia.

The participants in our study described how supportive performance reviews could lead to performance improvement for employees with dyslexia, particularly when employees felt safe to disclose. This aligns with work undertaken by von Scharder et al. (2014) [74] that suggested workplace improvements occurred when employees could identify that their difficulties were linked to their disability, could discuss this during their performance reviews, and felt supported to request reasonable accommodations in response to their disability.

### 5.2. Disclosure of Dyslexia by Employees

Disclosure of dyslexia in the workplace was a key issue explored in this research. Consistent with the literature, we found that disability disclosure in the workplace could also be met with negative consequences for employees, such as low supervisor expectations, isolation and exclusion from co-workers, stigmatisation, discrimination, impact on career progression and termination of employment [74,75,76,77].

The literature suggests that workplace culture plays a critical role in disclosure decision-making [74,77]. Our study confirmed that when managers were empathetic and understanding, staff felt safe to disclose their dyslexia, and this resulted in positive outcomes for both the employee and the employer. However, in our study, it was to the responsibility of the employee to self-identify or disclose to their managers that they had dyslexia. This reflects the findings of O’Dwyer and Thorpe (2013) [38]. To counter this, workplaces need to be upskilled to support employees with dyslexia so they can feel psychologically safe to disclose and then access the reasonable adjustments needed to perform at their best.

The participants in our study identified several barriers that might prevent employees from disclosing dyslexia. At the organisation level, these barriers included a lack of awareness of dyslexia, minimal or no training opportunities to manage and support dyslexic workers, and an absence of workplace disability policies and/or procedures. This was a similar finding to O’Dwyer and Thorpe (2013) [38], who found that within the Further Education sector, there was a focus on supporting the student who has dyslexia directly, rather than supporting or upskilling teachers to work effectively with dyslexic students. In addition, Further Education policies and procedures and consultants and external training bodies did not appear to inform employees or guide managers on how to support any teacher who was dyslexic.

All participants in this study had prior experience of dyslexia through family members or managing previous staff. However, participants noted there was a possibility that managers and employers without this prior experience might feel ill-equipped, overwhelmed, or susceptible to inadvertent discrimination when working with dyslexic staff. This supported the findings of O’Dwyer and Thorpe (2013) [38], who identified that managers might feel ill-prepared to provide the necessary support to a staff member who disclosed it. In light of more inclusive and diverse employee profiles within the workplace and the emergence of initiatives designed to increase the number of people with disabilities in the labour market, employers would be well-advised to foster a workplace that encourages disclosure and reduce the likelihood of negative consequences for the employee with dyslexia.

### 5.3. Discrimination of Dyslexic Employees

Despite the high global prevalence, there is still a profound lack of awareness and understanding of dyslexia at a societal level [30,31,78,79,80]. This lack of awareness can have a flow-on effect within the workplace, leading to dyslexia discrimination. In this study, participants acknowledged that those with dyslexia could be discriminated against across the employment lifecycle, starting at recruitment (e.g., during psychometric assessments and interview processes) and leading into onboarding and retention. This correlates with the literature, which describe perceived discrimination or unfair treatment during screening and recruitment process as those with dyslexia may not fit the job-profiles advertised. Additionally, the tools used to screen potential candidate may not accurately and equitably evaluate the capabilities and abilities of people with dyslexia [30,41,81,82]. Rao’s work demonstrated concerns raised by employers when looking to recruit individuals with disabilities such as neurodiverse people including the need to have in depth knowledge of multiple disabilities, the skills to manage mental health illnesses and training of staff in these disabilities [82]. Although research shows that neurodiverse employees are manageable and the return on invest (ROI) within this cohort is significant [82,83,84].

The participants in this study had clear, informal support systems in place to support their dyslexic employees. Yet, participants in some cases still felt that discrimination and stigma remained attached to the term ‘dyslexia’. This correlates with other studies that have found those with disabilities, including dyslexia, are often subjected to negative perceptions and stigmatisation or disparate treatment, including wage disparities [31,85,86,87,88,89,90]. Local research has shown that when employees with dyslexia feel they have insecure employment and unsupportive working conditions, they are at risk of poor mental health, well-being and early job burnout [30].

### 5.4. Creating Inclusive Workplaces

Despite the high prevalence rates of dyslexia in the workforce, our findings support the literature that indicates workplaces appear to lack the skills to support the dyslexic population as they transition from education into the workforce [38,39,41,91]. Our work adds to previous research by O’Dwyer and Thorpe (2013) [38] and Thorpe and Burns (2016) [39], who investigated managers’ understanding of dyslexia within the teaching population. Together, their work identified there was a lack of awareness, training, and support for managers of dyslexic staff and that dyslexia was still an invisible disability in the workforce. In these studies, it was reported that it was the responsibility of dyslexic employees to disclose and self-advocate for what they needed to survive in the workplace [31,38,39]. Due to the complex nature of dyslexia, employers and managers can face unique challenges and tension in the workplace that they may not be adequately prepared for or skilled to manage. 

These concerns are corroborated in a qualitative case study by Feggins (2022) [92], which concluded that managers, supervisors, and HR professionals need dyslexia awareness and education training, to better understand and support employees with disabilities. Feggins acknowledged the necessity for disability training tailored to Human Resource (HR) professionals and those who have responsibilities in the recruitment process. Alongside HR training, Feggins suggested disability awareness training for organisations emphasising diversity and inclusion [92]. Similarly, research undertaken by Wissell et al. (2022) [30] highlighted the need for dyslexia awareness training across the whole of an organisation, with specific training for HR and those in leadership roles (managers, team leaders and/or supervisors).

Our study indicated that managers believed reasonable adjustments were a key enabler to empowering their employees with dyslexia to succeed. However, formalised reasonable adjustments were not readily available to our sample, rather, this process was left to the discretion of managers. This reflects work by Wissell et al. (2022) [30], who also found that access to reasonable adjustments was the responsibility of managers and that managers often felt ill-equipped to support such requests. As Sumner and Brown (2015) [41] indicated, employers and managers require training to understand the possible accommodations that dyslexics may need and to determine whether these accommodations are reasonable, fair, justified, and of reasonable cost to the organisation.

Finally, and most importantly, our study indicates that when employers and managers have been previously exposed to, or had experience with dyslexia, they are more likely to make workplace adjustments for dyslexic employees, leading to more inclusive working environments. This emphasises the need for organisations to be proactive rather than reactive in their approach to supporting the specific needs of dyslexic staff, by facilitating awareness training for managers, leaders and HR staff, creating workplaces that feel psychologically safe to disclose, and providing access to reasonable work adjustments.

### 5.5. Implementation of International and National Policy Change That Can Create Workplace Inclusion

As signatories of the UNCRPD and Article 27 in particular, Australia has a responsibility to actively raising awareness, reduce discrimination and ensure the rights of those with dyslexia. Australia must adhere to the UNCRPD, as a signatory as we have set goals and targets to reach, a process that is monitored and reported on every four years [42]. Yet, international research in relation to dyslexia in the workplace has revealed a tension between the development of a policy and the real-life application of that policy [38,90,93]. In Australia in recent years, several initiatives have been launched to increase employment of people with disabilities, such as the ‘Employ My Ability’ Disability Employment Strategy, which support employment of people with a disability-Inclusive workplace cultures were people with disability thrive in their careers [94]. Nevertheless, as is the case with other similar initiatives, the ‘Employ My Ability’ strategy seems to excludes the needs of individuals with ‘hidden’ learning disabilities such as dyslexia [94]. Our work illustrates that the exclusion of dyslexia from national strategies may contribute to employers and managers lacking the skills required to support dyslexic employees and lead diverse teams.

At a micro level the employment sector must adhere to and meet the requirements under the Discrimination, Human Rights and Fair Work Acts [46,47,48], through policies and policy enactment that raise awareness and disability awareness training, reduces discrimination through the adaptations to recruitment, onboarding and retention practices, ease of access to reasonable adjustments and psychologically safe workplaces could all help reduce the perceived risk of discrimination occuring and lend themselves to improving workplace practices. These strategies are enablers of a cultural movement of inclusion and diversity which is truly embraced in practical terms across the employment sector.

After observing the impact of the COVID-19 pandemic in Australia, we now know that workplaces can be adaptive and flexible. We also know that societal changes and challenges to workplace norms have led to a greater focus on inclusion and diversity within the workplace. These positive changes are a great start, but they now need to be broadened to embrace those with dyslexia and other ‘invisible’ disabilities so that all employees can be effective and productive contributors in the workplace.

## 6. Limitations

This preliminary study had several drawbacks that must be recognised. First, the cohort was very small, and the results cannot therefore be generalised with any confidence. This preliminary study could only recruit four participants, it could be assumed that uptake was low because employers and managers are not coming across employees with dyslexia because workplaces are not providing psychologically safe environment to disclose. Employers and managers may feel vulnerable to be open about how they are managing employees with dyslexia, and it may shine a light on gaps in their workplaces that they do not want to disclose, or alternatively workplaces just have such little understanding and awareness of dyslexia that employers do not believe they have dyslexic employees, and they are performance managing those with dyslexia out.

Further, our participants were weighted towards those with previous lived experiences of dyslexia, which may bias the results. The challenges and achievements presented here may not signify the wider experiences of leaders who have worked with dyslexic employees. The personal experiences and knowledge of Author One, who is dyslexic, may have influenced data collection. Future research should consider involving a broader range of leaders that work with dyslexic employees.

## 7. Conclusions

Currently there is limited research of a similar nature that has investigated the experiences of employers and managers of dyslexic employees. In this preliminary study although the sample size was small, we found that our participants’ workplaces appeared to be ill-equipped to support dyslexic employees across the workplace ecosystem; it was only when leaders had pre-exposure to dyslexia that they knew how to effectively influence support to assist their dyslexic employees. Overwhelmingly, leaders felt that when dyslexic employees had access to appropriate reasonable adjustments, they made a significant and positive contribution to their workplaces which could lead to improved economic outcomes for employers and society in general.

Generally, the literature on dyslexia in the workplace is weighted to the experiences reported by employees with dyslexia. Further research from a management point of view would assist in developing support strategies for dyslexic employees across various industries and roles. Discouragingly, there is a lack of awareness of dyslexia within workplaces and society. Yet, we found that when empathy was paired with an awareness and understanding of dyslexia as part of the workplace culture, those with dyslexia were not only surviving, but also thriving.

## Figures and Tables

**Table 1 ijerph-19-11991-t001:** Participants’ Characteristics.

Participants	Gender	Company Size (Employees)	Industry	Length of Time in Current Role	Occupation	Participant Location (State)	Location of Business
P1	F	95	Hospitality and Tourism	5–10 years	Challenge and Culture Manager within an international chain hotel, oversaw staff across front office, food and beverage, finance, kitchen, and maintenance	New South Wales	Metro
P2	F	100	Non-Government Organisation	3–5 years	Associate Director working in an international not-for-profit organisation with an Australian branch	Victoria	National
P3	M	13	Graphic Design Agency	10+ years	Director and an Account Manager for their own national marketing and communication company, responsible for looking after projects and staff.	Victoria	Metro
P4	F	30,000	Manufacturing	18 months	Business Development Officer; had previously worked as a Human Resources Manager for two national companies in manufacturing.	Victoria	National

## Data Availability

The data that support the findings of this study are available from the corresponding author upon satisfactory request.

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
