# Peer review of "Leading Diverse Workforces: Perspectives from Managers and Employers about Dyslexic Employees in Australian Workplaces"

_ijerph, 2022, doi:10.3390/ijerph191911991_

Round 1
Reviewer 1 Report
Dear Authors,
I appreciate your effort to add scientific results to this neglected topic (the experiences of managers with dyslexia workers and the inclusion of adults with dyslexia in workplaces).
Your manuscript needs big changes because two reasons:
1. You must change the special issue. This belongs to the treatment of dyslexia, and in my reading, the problem formulation must be connected with CHILDREN having this problem. The company leaders' experiences with dyslexia employees should be included in another topic. My recommendation is the Special Issue "Current Insights in Promoting Well-Being at Work", in the same journal.
2. The scientific methodology of the manuscript is very low. My recommendation is to add more interviews with managers (more than ten) and the analysis must be more clear. I attached the manuscript with my comments.

Author Response
Reviewer Abstract too long
Please see amended abstract below
Background: Dyslexia is a specific learning disability affecting around 1 in 10 Australian adults. It presents unique challenges for employees in the workforce, yet community and workplace awareness of the challenges of dyslexia is limited. The aim of this preliminary research was to explore the experiences and perspectives of Australian employers and managers responsible for supervising employees with dyslexia in the workplace.
Materials and Methods: Using a qualitative research design, we conducted in-depth interviews with four managers who had current or previous experience managing employees with dyslexia. We used a deductive approach to analyse the data and categorise responses to the study questions.
Results: Participant responses indicated that there is a lack of awareness and understanding of dyslexia within Australian workplaces. Participants identified challenges facing employees with dyslexia in the workplace including: differing personal levels of confidence and comfort in disclosing disability; the possibility of discrimination, and a lack of inclusive organisational practices and processes. Suggestions for ways to improve workplaces for dyslexic employees included: additional support for leaders and managers to drive inclusive leadership, and additional training for leaders and managers on how to best support employees with dyslexia.
Conclusions: While only a small sample size, this study indicates that further research is needed to better understand the working environment of Australian leaders and managers. It appears that leaders and mangers need skills and knowledge to better support employees with dyslexia and in doing so create more inclusive workplaces.
Point 1: You must change the special issue. This belongs to the treatment of dyslexia, and in my reading, the problem formulation must be connected with CHILDREN having this problem. The company leaders' experiences with dyslexia employees should be included in another topic. My recommendation is the Special Issue "Current Insights in Promoting Well-Being at Work", in the same journal.
Response: Thank you for your recommendation, we will take this up with the editor.
Point 2: What are the research questions?
Response: The research question has been included in the abstract as suggested
Point 3: Prevalence rates where do they relate to:
Response: The word global has been added to the prevalence rates as outlined below:
The reported global prevalence of dyslexia can therefore range from 5% to 20%.
Point 4: What are the other learning disabilities.
Response: An explanation ash been included in the background as outlined below:
Approximately 3–7% of the population have dyscalculia (Butterworth, 2019; Glazzard & Dale, 2013; Haberstroh & Schulte-Korne, 2019; Kucian & Aster, 2015; Wilson et al., 2015) have severe difficulties performing arithmetic calculations that persist into adulthood (Butterworth, 2019; Wilson et al., 2015). While approximately 3-15% of the population have dysgraphia, a specific learning disorder affecting the written expression of symbols and words that persist into adulthood (Döhla & Heim, 2015; McBride, 2019; McCloskey & Rapp, 2017).
Point 5: The scientific methodology of the manuscript is very low. My recommendation is to add more interviews with managers (more than ten) and the analysis must be more clear. I attached the manuscript with my comments.
Response: Unfortunate we were unable to recruit more than four participants into this study. To address this we have stated that this is a preliminary study in the aim, modified the methods section and added further details to the limitation section as please see below.
Aim: This is a preliminary study that forms part of a larger research project that investigated the lived experiences of adults with dyslexia in Australia, focusing on the workplace.
Methods: Unlike quantitative research, qualitative research gathers data until saturation is reached, and the collected data tell a rich, complex, and in-depth story about the topic under examination. Compared to quantitative research, sample sizes for qualitative research are typically much smaller (Dworkin, 2012; Malterud et al., 2016). It has been suggested in a large number of articles, book chapters, and books that anywhere from five to fifty participants is sufficient (Dworkin, 2012).
Limitations: This preliminary study had several drawbacks that must be recognised. First, the cohort was very small, and the results cannot therefore be generalised with any confidence. Finally, this preliminary study could only recruit four participants, it could be assumed that uptake was low because employers and managers are not coming across employees with dyslexia because workplaces are not providing psychologically safe environment to disclose. Employers and managers may feel vulnerable to be open about how they are managing employees with dyslexia and it may shine a light on gaps in their workplaces that they do not want to disclose, or alternatively workplaces just have such little understanding and awareness of dyslexia that employers don’t believe they have dyslexic employees and they are performance managing those with dyslexia out.
Point 6: Concerns my knowledge your approach should be inductive (qualitative measure in exploratory research) Or test a theory or hypothesis
Response: We have further explained why we think this research is deductive rather than inductive this has been added to the manuscript, please see below:
We choose to use a deductive content analysis approach using a set of pre-defined questions conducted through semi-structured, in-depth interviews to collect substantial responses from participants enabling us to draw specific conclusions from the data. Further follow-up and clarifying questions emerged during interviews with participants.
Point 7: What are the connections between them? Problem formulations is very important here.
Response: Thank you for your comments we have added additional detail in the background as detailed below:
However, despite the rollout of progressive international and national strategies (Commonwealth of Australia, 2021b, 2022) and changes to legislation (Australia, 1992, 2009) individuals with dyslexia have been left behind in particular within the employment sector as highlighted by research undertaken by Wissell et al., (2022) who found those with dyslexia faced a significant number of barriers to succeeding in the workplace despite these legislations and national strategies. In fact, those with learning disabilities have been completely excluded from the National Disability Insurance Scheme (NDIS) and the Medicare Benefit Scheme (MBS) an Australia medical healthcare services subside scheme (Commonwealth of Australia, 2022).
Point 8: Develop the strategy on the introduction in conclusion
Response: Thank you for your comments we have added additional detail in the body of the discussion and made the conclusion more concise:
5 Implementation of international and national policy change that can create workplace inclusion
As signatories of the UNCRPD and in particular Article 27, Australia has a responsibility to actively raising awareness, reduce discrimination ensure the rights of those with dyslexia. Australia must adherence to the UNCRPD, as a signatory as we have set goals and targets to reach, and this process is monitored and reported on every four years [42]. Yet international research in relation to dyslexia in the workplace has revealed a tension between the development of a policy and the real-life application of that policy [38, 91, 94]. In Australia in recent years, several initiatives have been launched to increase employment of people with disabilities, such as the ‘Employ My Ability’ Disability Employment Strategy, which support employment of people with a disability – Inclusive workplace cultures were people with disability thrive in their careers [95]. Nevertheless, as is the case with other similar initiatives, the ‘Employ My Ability’ strategy seems to excludes the needs of individuals with ‘hidden’ learning disabilities such as dyslexia [95]. Our work illustrates that the exclusion of dyslexia from national strategies may contribute to employers and managers lacking the skills required to support dyslexic employees and lead diverse teams.
At a micro level the employment sector must adhere to and meet the requirements under the Discrimination, Human Rights and Fair Work Acts [47-49], through policies and policy enactment that raise awareness and disability awareness training, reduces discrimination through the adaptations to recruitment, onboarding and retention practices, ease of access to reasonable adjustments and psychologically safe workplaces could all help reduce the perceived risk of discrimination occuring and lend themselves to improving workplace practices. These strategies are enablers of a cultural movement of inclusion and diversity which is truly embraced in practical terms across the employment sector.
After observing the impact of the COVID-19 pandemic in Australia, we now know that workplaces can be adaptive and flexible. We also know that societal changes and challenges to workplace norms have led to a greater focus on inclusion and diversity within the workplace. These positive changes are a great start, but they now need to be broadened to embrace those with dyslexia and other ‘invisible’ disabilities so that all employees can be effective and productive contributors in the workplace.

Reviewer 2 Report
Although this study is based on a very small sample, it provides an interesting insight on employers´ perceptions about dyslexia. I will only do three small suggestions to improve the paper:
1. The first suggestion has to do with the structure of the paper, which has to be improved: it has to be organized in sections and subsections, which should be numbered. The paper has 4 main sections, the titles o which should be: Section 1. Introduction; Section 2. Methodology; Section 3. Discussion; and Section 4. Conclusions. Inside each section, a clearer organization of subsections and paragraphs (also numbered) is needed.
2. Concerning the background, I think some reference should be made to the UN Convention on the Rights of Persons with Disabilities. The authors do refer to the Australian Disability Strategy and to some other national regulations, but surprisingly they do not mention the UN Convention, which is nowadays the most important normative framework for disability policies, which is legally binding also in Australia and which devotes a long and important article (article 27) to the right to work and employment.
3. I think the conclusions should be enriched with some policy recommendations based on the study. There is indeed one suggestion from the authors (to incorporate learning disabilities to national disability policies and strategies), but I think this last paragraph of the conclusions could be developed, adding some opinion for example on what can be done to prevent discrimination of people with dyslexia or to provide these persons an appropriate support at work place.
Author Response
Point 1 The first suggestion has to do with the structure of the paper, which has to be improved: it has to be organized in sections and subsections, which should be numbered. The paper has 4 main sections, the titles o which should be: Section 1. Introduction; Section 2. Methodology; Section 3. Discussion; and Section 4. Conclusions. Inside each section, a clearer organization of subsections and paragraphs (also numbered) is needed.
Response: Thank you for your recommendation, the manuscript has now been updated to include numbering and subtitles.
Point 2 Concerning the background, I think some reference should be made to the UN Convention on the Rights of Persons with Disabilities. The authors do refer to the Australian Disability Strategy and to some other national regulations, but surprisingly they do not mention the UN Convention, which is nowadays the most important normative framework for disability policies, which is legally binding also in Australia and which devotes a long and important article (article 27) to the right to work and employment.
Response: Thank you for your recommendation, the manuscript has now been updated to include reference to the UN Convention on the Rights of Persons with Disability in 2006 and article 27 as outlined below:
Recently the term ‘disability’ has also broadened to encompass both visible (e.g., physical limitations) and invisible (e.g. chronic fatigue, attention deficit hyperactivity disorder) conditions (Doyle & McDowell, 2020; Sumner & Brown, 2015). Encouragingly, along with the broadening of the term ‘disability’ there has been a cultural shift in global and national policy regarding the need to increase community awareness of disability and inclusion as seen through the United Nations Convention on the Rights of Persons with Disability in 2006 which Australia is a signatory of, and as such has a responsibility to protect the human rights and inherent dignity of persons with disabilities (McCallum, 2020). More poignantly, Article 27- Work and employment that recognised the right of persons with disabilities to work, on an equal basis with others; this includes the right to the opportunity to gain a living by work freely chosen or accepted in a labour market and work environment that is open, inclusive and accessible to persons with disabilities (Nations, 2006). At a national level Australians have seen initiative such as the Australia’s National Disability Strategy, implemented in 2010 (Commonwealth of Australia, 2021a) and expanded in 2021 and more recently the National Disability Insurance Scheme, which provides support to people with intellectual, physical, sensory, cognitive and psychosocial disability (Commonwealth of Australia, 2022). The initiatives outline Australia’s national aspirations to respect inclusivity and equality and outlines state government and ‘whole of community’ responsibilities (Commonwealth of Australia, 2021b). Change towards greater inclusivity is also endorsed in legislative requirements such as the Discrimination, Equality and Fair Work Acts (Australia, 1992, 2009, 2010).
Point 3. I think the conclusions should be enriched with some policy recommendations based on the study. There is indeed one suggestion from the authors (to incorporate learning disabilities to national disability policies and strategies), but I think this last paragraph of the conclusions could be developed, adding some opinion for example on what can be done to prevent discrimination of people with dyslexia or to provide these persons an appropriate support at work place.
Response: Thank you for your comments we have added additional detail in the discussion section
As signatories of the UNCRPD and in particular Article 27, Australia has a responsibility to actively raising awareness, reduce discrimination ensure the rights of those with dyslexia. Australia must adherence to the UNCRPD, as a signatory as we have set goals and targets to reach, and this process is monitored and reported on every four years [42]. Yet international research in relation to dyslexia in the workplace has revealed a tension between the development of a policy and the real-life application of that policy (Gates, 2010; O'Dwyer & Thorpe, 2013; Scott, 2016). In Australia in recent years, several initiatives have been launched to increase employment of people with disabilities, such as the ‘Employ My Ability’ Disability Employment Strategy, which support employment of people with a disability – Inclusive workplace cultures were people with disability thrive in their careers [95]. Nevertheless, as is the case with other similar initiatives, the ‘Employ My Ability’ strategy excludes the needs of individuals with ‘hidden’ learning disabilities such as dyslexia [95]. Our work illustrates that the exclusion of dyslexia from national strategies can contribute to employers and managers lacking the skills required to support dyslexic employees and lead diverse teams.
At a micro level the employment sector must adhere to and meet the requirements under the Discrimination, Human Rights and Fair Work Acts (Australia, 1992, 2009, 2010), through policies and policy enactment that raise awareness and disability awareness training, reduces discrimination through the adaptations to recruitment, onboarding and retention practices, ease of access to reasonable adjustments and psychologically safe workplaces would reduce the risk of discrimination occuring and lend themselves to improving workplace practices. These strategies are enablers of a cultural movement of inclusion and diversity which is truly embraced in practical terms across the employment sector.
After observing the impact of the COVID-19 pandemic in Australia, we now know that workplaces can be adaptive and flexible. We also know that societal changes and challenges to workplace norms have led to a greater focus on inclusion and diversity within the workplace. These positive changes are a great start, but they now need to be broadened to embrace those with dyslexia and other ‘invisible’ disabilities so that all employees can be effective and productive contributors in the workplace.

Round 2
Reviewer 1 Report
Dear Authors,
the quality of the manuscript is much better.
I recommend using the template of the journal and reviewing the numbers and the style in the bibliography.
I wish you all the best!